# Different pedagogical approaches to motor imagery both demonstrate individualized movement patterns to achieve improved performance outcomes when learning a complex motor skill

**Riki S. Lindsay**[1]\*, **John Komar**[2], **Jia Yi Chow**[2], **Paul Larkin**[1], **Michael Spittle**[1]

**1** Institute for Health and Sport, Victoria University, Melbourne, Australia, **2** National Institute of Education, Nanyang Technological University, Singapore, Singapore

\* r.lindsay@federation.edu.au

**Data Availability Statement:** All relevant data are within the manuscript and its Supporting Information files.

## Abstract

Cognitive training techniques such as motor imagery (MI)–cognitive simulation of movement, has been found to successfully facilitate skill acquisition. The MI literature emphasizes the need to accurately imitate key elements of motor execution to facilitate improved performance outcomes. However, there is a scarcity of MI research investigating how contemporary approaches to motor learning, such as nonlinear pedagogy (NLP), can be integrated into MI practice. Grounded in an ecological dynamics approach to human movement, NLP proposes that skilled action is an emergent process that results from continuous interactions between perceptual information of the environment and movement. This emergent process can be facilitated by the manipulation of key task constraints that aim to encourage learners to explore movement solutions that satisfy individual constraints (e.g., height and weight) and achieve successful performance outcomes. The aim of the present study was to explore the application of a NLP approach to MI approach for skill acquisition. Fourteen weightlifting beginners (two female and 12 male) participated in a 4-week intervention involving either NLP (i.e. analogy-based instructions and manipulation of task constraints) or a linear pedagogy (LP; prescriptive instructions of optimal technique, repetition of same movement form) to learn a complex weightlifting derivative. Performance accuracy, movement criterion (barbell trajectory type), kinematic data, and quantity of exploration/exploitation were measured pre-mid-post intervention. No significant differences (p = .438) were observed in the amount of exploration between LP (EER = 0.41) and NLP (EER = 0.26) conditions. Equivalent changes in rearward displacement (R×D) were observed with no significant differences between conditions for technique assessments 1, 2, or 3 (p = .13 - .67). Both NLP and LP conditions were found to primarily demonstrate 'sub-optimal' type 3 barbell trajectories (NLP = 72%; LP = 54%). These results suggest that MI instructions prescribing a specific movement form (i.e., LP condition) are ineffective in restricting available movements to a prescribed technique but rather the inherent task constraints appear to 'force' learners to explore alternative movement solutions to achieve successful performance outcomes.

**Funding:** The author(s) received no specific funding for this work.

**Competing interests:** The authors have declared that no competing interests exist.

Although MI instructions prescribing specific techniques have previously supported improved skill development, the current findings indicate that learners may self-organise their movements regardless of MI instructions to satisfy individual and task constraints while achieving improved performance. Therefore, it may be beneficial to consider scripts that are more outcome focused and incorporate task constraints to facilitate learners' inherent exploration of individual task solutions.

## Introduction

Motor learning is often defined as a relatively permanent change in skill performance due to physical practice of a movement [1], research, however, highlights that cognitive training techniques, such as mental imagery, can also facilitate motor learning when combined with physical practice or alone [2–4]. Mental imagery refers to the ability to simulate perceptual and motor information in our mind without sensorimotor input [5]. Under the broad category of mental imagery is motor imagery (MI), which focuses specifically on the cognitive recreation of a movement without overt motor execution [6]. The mechanisms underlying MI have typically been aligned with an information processing-based view of cognition, whereby MI stores and retrieves internal movement representations from long-term memory [7, 8]. Grounded in an information-processing view, motor simulation theory (MST) proposes that the observed effects of MU are the result of shared neural mechanisms to those utilized during actual motor execution, termed the *functional equivalence* hypothesis [9, 10], whereby MI and overt movement are functionally equivalent due to a shared mental representational system involved in creating motor actions [11]. For example, studies have indicated substantial overlap of neural activity in motor and premotor areas (cerebellum, inferior frontal gyrus, and ventrolateral thalamus) during MI and motor execution [12, 13]. MI practice has also been shown to elicit training-related adaptations in central neural structures, such as the corticospinal pathway, like what is observed in physical training. For example, Leung, Spittle [14] compared the effect of physical training against MI training alone for a bicep-curl strength exercise. Following a three-week intervention period, both conditions demonstrated significant increases in strength (i.e., one-repetition maximum), coupled with equivalent increases in corticospinal excitability. Taken together, these findings suggest that MI and movement execution not only activate similar neural structures, but can produce similar training-related cortical adaptations [14, 15].

Drawing on MST and empirical neurophysiological findings, MI training has primarily aligned with cognitive views of motor learning, in which skilled action is acquired through the development of internal representations that lead to changes in cognitive processes that drive motor execution [16]. Subsequently, MI training is often designed to reproduce and strengthen specific internal movement representations with the assumption that it can be transferred to range of movement contexts [1, 16]. Empirical findings appear to support this view, with MI training being found to produce both significant changes in mental representations and performance outcomes, indicating that skill performance and development may be predicated on the acquisition, storage and updating of internal movement representations [17–19]. In contrast to cognitive views, action-based theories of cognition emphasize the emergent nature of psychological states [20]. Specifically, an ecological dynamics view suggests that skilled action is a continuous and mutually influencing process, with the purpose being to hone an individual's ability to perceive and pick up information in an everchanging environment to inform successful performance outcomes [21, 22].

Contemporary approaches to motor learning, such as Nonlinear Pedagogy (NLP), draw on the ecological dynamics view to provide a skill development framework that acknowledges the emergent nature of skilled action. From a NLP perspective, the aim is to develop an individual's ability (i.e., performer constraints) to successfully adapt their movements to changing environmental constraints. Therefore, behavioral flexibility is a key attribute of skilled action, in which an individual can explore possible movements and attune to relevant information in satisfying the overall task goal [23, 24]. Individuals adapt to environmental perturbations by exploring alternative movement options to meet the changing task demands. For example, Müller et al. [25] showed that expert cricket batsmen practicing under occluded and normal vision conditions demonstrated unique coordination patterns, yet performance outcomes were equivalent between all batsmen. These findings suggest that the aim of developing expertise may not be to replicate an 'optimal' mental model, but rather create an ability to adapt and produce stable individualized movements in the face of a dynamic performance environment [26]. Similarly, Lindsay, Oldham [27] noted that after 6-weeks of MI practice, power clean barbell trajectories were highly individualized in novice lifters, suggesting the need for further research to investigate the influence of movement variability in MI practice. These findings suggest that an alternative approach to skill acquisition may be a fruitful line of enquiry to contribute to our present understanding of how MI interventions can cater for individual factors that influence skill development.

Subsequently, skill acquisition may be considered more appropriately as *skill adaption*. This change in definition shifts the aim of practice from attaining an 'optimal' technique to providing opportunities for learners to explore and exploit the perceptual-motor workspace, facilitating the development of stable and adaptable coordination solutions [21]. The continuous development of the perceptual-motor workspace creates new coordination possibilities that can be explored [28]. The exploration process can be formalized through the measurement of variability and defined broadly as the engagement in a range of different coordination solutions to arrive at a specific task goal [29, 30]. By contrast, exploitation involves consecutive reproduction of the same coordination pattern, facilitating behavior stabilization. Captured this way, skilled action is developed via a back-and-forth process of exploration, compilation, and stabilization of coordination patterns that can adapt under dynamic conditions [29].

To facilitate the development of adaptable movement skills, NLP advocates for the following practice principles; (1) representative practice simulations to performance situations that present critical aspects of competitive environments; (2) careful and considered manipulation of task/environmental constraints (e.g., playing surface, number of players, size of the field) to facilitate exploration and exploitation of perceptual-motor workspace; (3) leveraging variability in practice to encourage adaptive and exploratory behavior, guiding the learner to explore individually relevant and appropriate performance solutions; and (4) implement instructions that encourage processes of self-organization by focusing attention on movement outcomes as opposed to specific body positions(i.e. internal focus) [30].

Evidence indicates that NLP informed practice of open skills facilitates exploratory behaviour (i.e., movement variability) during learning without negatively impacting performance [31]. For example, Lee, Chow [31] demonstrated that novice learners practicing a tennis skill under NLP displayed greater exploratory behavior than linear pedagogy (LP; repetitive practice of 'optimal' technique), even though both groups displayed similar performance improvements. These findings indicate that adherence to an 'optimal' technical model does not ensure superior performance and though commonly viewed as 'errors', exploration/movement variability could play a functional role in facilitating the development of individualized performance solutions. Pertinent to the design of MI practice, NLP highlights the importance of manipulating task constraints to encourage exploration and facilitate the development of

adaptable, individualized movement solutions [26]. Therefore, the aim of a NLP informed MI instructions would be to describe critical aspects of the learning environment, such as task constraints (e.g., barrier in front of someone lifting a barbell), rather than presenting a description of the 'optimal' technique. Presently, no studies have formally assessed the influence of a NLP informed MI approach to skill development.

The present study aimed to explore the application of a NLP informed MI practice in relation to a traditional linear style of MI practice for beginners learning a movement form-based skill, a weightlifting skill known as the power clean (PC). It was hypothesized that: 1) the linear style of practice would develop a higher frequency of 'optimal' movement patterns; 2) modification of task constraints in NLP condition would help facilitate exploratory behavior and guide learners toward performance relevant solutions; and 3) both conditions would demonstrate the same levels of performance accuracy, as measured by the distance the barbell travels forward ($F \subseteq D$) and backward ($R \subseteq D$) relative to the start position.

## Materials and methods

### Participants

Sixteen healthy adult participants (3 female; 13 males) agreed to participate in the study. Due to personal reasons, two participants did not complete the study, leaving a total of fourteen participants (2 female, 12 male; 29.1±3.3 years). Therefore, participants were not randomly assigned to the LP or NLP conditions to maintain balanced groupings. Participants were reimbursed $100AUD in the form of supermarket vouchers for travel, parking expenses and time. All were healthy and free of acute/chronic injuries and provided written, informed consent. All participants had less than three months of formal experience learning the power clean movement and two years of general gym training experience [32]. Based on these criteria, participants were naïve to the proposed motor skill to be learned and considered beginners [32, 33], conforming to the control stage of motor learning [28]. The university ethics committee approved the present study.

### Procedure

The present study comprised of a pre-intervention technique assessment, followed by a 4-week intervention (eight MI sessions, each lasting 30 minutes), a mid-intervention (end of week 2) and post-intervention technique assessment approximately 24-hours after the intervention.

### Technique assessment procedures

Prior to the commencement of the pre-intervention technique assessment, all participants completed the Movement Imagery Questionnaire-Revised (MIQ-R; [34]) to determine their ability to perform MI before beginning MI-based practice. The MIQ-R comprises eight items that aim to assess visual and kinesthetic imagery ability (four items for each domain). Participants were required to imagine four different movements visually or kinesthetically. After completing each movement, participants used a seven-point Likert scale (1 = very difficult to see or feel; 7 = very easy to see or feel) to rate their imagery performance. The ability to perform MI was based on attaining an average score above 4 (Neutral, not easy, not hard to see or feel) (16). The MIQ-R has high internal (visual subscale = 0.84; kinesthetic subscale = 0.88) and test-retest reliability (visual subscale = 0.80; kinesthetic subscale = 0.88) [35].

Following completion of the MIQ-R a total of forty-eight reflective markers with 14mm were attached to upper and lower body landmarks according to the Plug-In-Gait model (Fig 1). Two

reflective markers were also placed on the right and left side of the barbell to trace the trajectory [36]. Markers were needed to construct a 3-D model to extract kinematic movement data.

Prior to the beginning of the pre-intervention session participants were provided with a demonstration by an experienced international level coach (five years coaching and teaching experience, including at international competitions) of the PC movement. This was due to participants being at a beginner level to reduce the risk of injury. Following the demonstration, a standardized warmup of 5 trials with an empty barbell, followed by 3⊆5 repetitions up to a total weight of 30kg. The mid (approximately 24 hours after MI session 4) and post-intervention (approximately 24 hours after MI session 8) technique assessments comprised of a standardized warmup (5 trials with an empty barbell), followed by 3⊆5 trials with a total weight of 30kg. Observations from pilot data indicated that 30kg was an appropriate resistance level to limit the risk of injury for beginners. Following each set, participants were required to rest for 2–5 minutes to reduce the effects of fatigue.

## Intervention

Following the pre-intervention technique assessment, the participants completed eight MI practice sessions (approximately 30 minutes), across a 4-week intervention period, to learn the PC using either a NLP or LP informed approach. For both conditions guidelines from the PETTLEP framework were followed to replicate elements of the performance environment as closely as possible [37]. Therefore, participants were instructed to wear the same clothing and footwear they would use when usually performing the movement task and were physically standing in front of a barbell loaded with 30kg in a gym environment congruent with where the movement is usually performed. Prior to each session participants were guided through a standard physical warmup routine to raise the heart rate and psychologically prepare participants to engage in the session. When in the start position of the movement, participants would listen to either a LP or NLP constructed audio recorded script that guided them through 3⊆5 MI trials. After each MI trial was completed, participants were required to signal to the researcher that they had completed a trial. This meant that the volume of training could be accurately accounted for with both conditions completing 120 MI trials over 4 weeks. This was implemented to ensure the correct volume of practice was being completed. The intervention was developed by five academics knowledgeable in MI, NLP and LP, and Olympic weightlifting respectively. Both NLP and LP interventions were delivered by the same researcher based on the methodology constructed prior to the beginning of the intervention.

Both NLP and LP instructions were designed on the understanding that when performing the PC movement there is a heavy reliance on proprioceptive sensory information to regulate movement posture and control, as individuals performing these movements need to approximate their body positions spatially by "feeling" the movement as opposed to "seeing" themselves performing it [38]. Therefore, kinesthetic MI was the primary form of MI for practice in the present study. Kinesthetic MI aims to elicit sensory aspects of the motor task from the first-person perspective, primarily focusing on the feel and timing of the action [39]. The audio recording also incorporated the initial visual aspects related to the physical movement, directing participants to focus on a specific point in front of them before performing the movement. For the present study, examples of the instructions given to participants included details such as "feel the rough grip of the bar as it sits in your hands" and "explosively shrug your shoulders".

For the NLP condition, instructions were analogy-based (i.e., focused on the movement outcome) to encourage self-organization processes and limit conscious movement control, aligning with key NLP principles of practice design [30, 40]. This included MI instructions

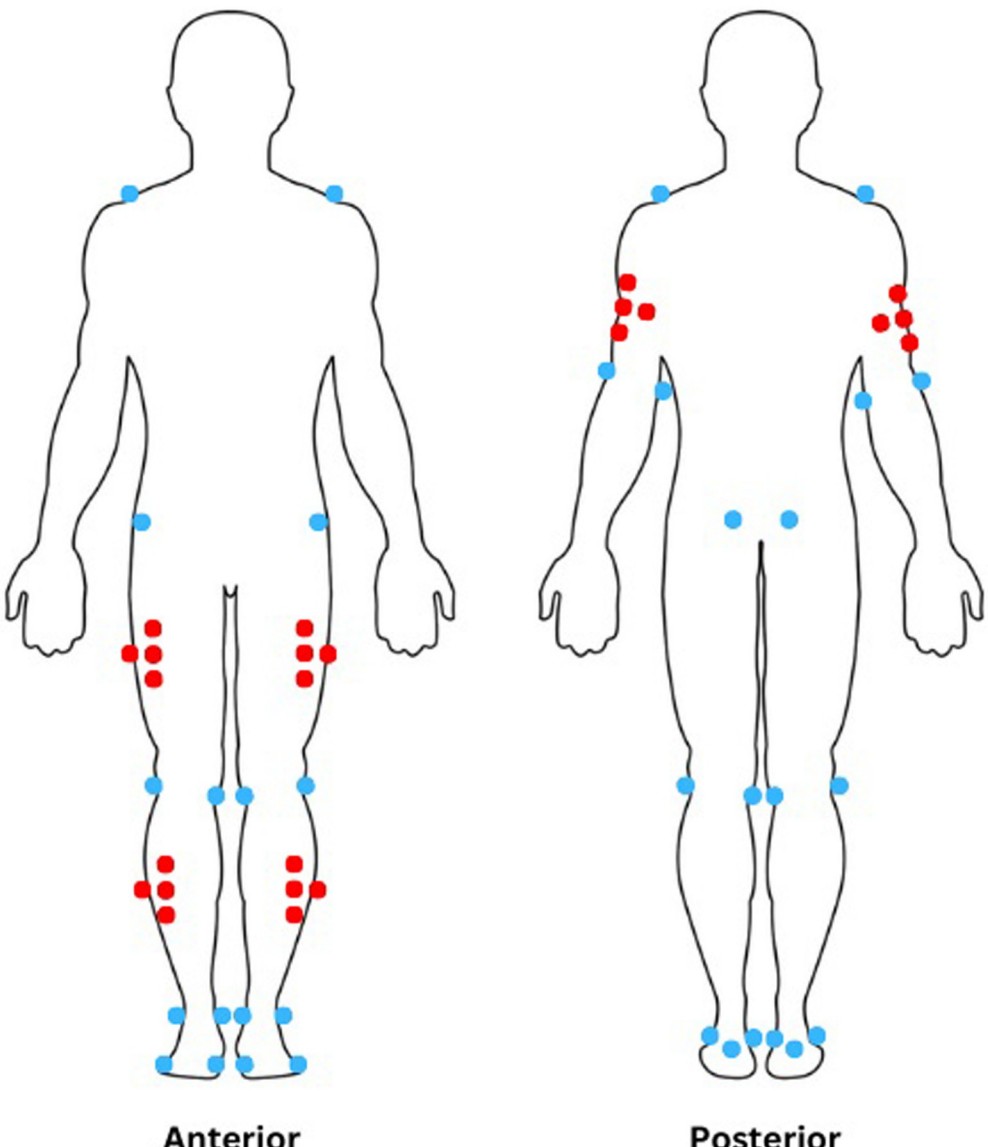

**Fig 1. Position of reflective markers.** Red markers were used to track segments. Blue markers denote selected anatomical landmarks.

such as "try and flick the bottom of your shirt as you pull upwards" and "explode upwards like you are jumping straight up". Manipulation of constraints are informed by principles of NLP, such as task constraints, aim to encourage exploration of individualized movement solutions. The manipulation of task constraints in NLP instructions included chalk on the barbell and poles in front of the barbell (Table 1). Participants were blinded to the true purpose of each constraint and were only told to either not hit the poles in front of them or try and leave a chalk mark on their thighs with the barbell [41]. The introduction of practice variability to encourage learner exploration is a key principle of NLP learning design [30]. Subsequently, constraints were described in MI instructions for the NLP condition to introduce participants to variability. In order not to overload participants in the early stages of learning, variability was introduced into MI practice between sessions 3–6 [26]. To replicate the physical practice

**Table 1. Summary of the constraints incorporated into NLP informed MI scripts.**

| Task constraint | Constraint details | Nonlinear Pedagogy design principle |
|---|---|---|
| Chalk on the barbell | Chalk was applied to the bar to encourage participants to pull the bar back towards the body during the lift and keep the bar in contact with the thighs while transitioning from below to above the knee and into the *second pull* position. If participants were keeping the bar in contact with the thighs, chalk from the bar would show where contact was occurring. | *Effective manipulation of tasks constraints.* The chalk on the barbell aimed to facilitate exploration and exploitation of alternative movement solutions, such as different starting heights of the *second pull* position. *Leveraging functional variability.* The chalk aimed to amplify exploration of different positions of the *second pull* to facilitate the emergence of individualized solutions during this phase of the lift. |
| Poles in front of participant | Two poles were placed in front of the participant to restrict forward movement of the bar. Participants would lift in front of the poles while trying to avoid contacting them. | *Reducing conscious control of the movement.* The poles aimed to focus attention on the movement outcome, to encourage self-organizing processes. *Leveraging functional variability.* By placing the poles in front of the learner this aimed to amplify exploratory activity and guide the learner toward performance solutions that matched specific capabilities, skill and experience. |

environment, constraints were physically present for the NLP condition, however, learners only imagined practicing under constraints, and these were not present for any testing sessions.

Conversely, the LP intervention was design on the understanding that during skill acquisition learners should be directed toward an 'optimal' technique and this is achieved through repetitive practice. In weightlifting research, the 'optimal' technical model is commonly described as a type one barbell trajectory [42]. The type one barbell trajectory displays limited forward movement and more rearward pulling of the barbell toward the body, meaning the barbell is caught closer to the lifter's base of support [43]. Therefore, LP instructions involved details of what is considered an 'optimal' PC technique (type one barbell trajectory) and were movement form orientated, aiming to have learners adopt a very specific movement form and leave little opportunity to explore alternate techniques [44]. The LP condition received prescriptive MI instructions according to different phases of the lift. For example, the second pull phase: "As your lower body extends forcing you to be right up on your toes" and the turnover of the barbell onto the shoulders: "Bend your elbows and pull your body under the bar". The PC instructions were developed and verified by an experienced weightlifting coach with international experience. MI instructions for the LP condition remained unchanged for the entire intervention. For a record of MI instructions for LP and NLP conditions, see S1 File.

## Apparatus and measurements

### Movement patterns

The 48 retroreflective markers fitted on predetermined anatomical landmarks by a 14-camera (T-series T40) motion capture system (Vicon Inc., Denver, Co, USA). Reconstructed trials were processed using Vicon Nexus software (2.10.1) and then analyzed using Visual 3D software (C-Motion Inc). Nine time-continuous kinematic variables were identified from previous research [45] and were computed in a local reference: right and left shoulder flexion/extension, abduction/adduction, pelvis flexion/extension, right and left knee flexion/extension, and left and right ankle flexion/extension. A low pass Butterworth digital filter at a frequency of 10Hz was used on all kinematic data, and filtered position data was time-normalized to 100 data

points to enable comparisons to be computed across trials and participants and cluster analysis (see section on data analysis). A static calibration trial was used to establish locally fixed segment coordinate systems and establish X, Y and Z axes.

## Performance accuracy: Horizontal barbell displacement

Performance accuracy was determined based on the overall distance the barbell travelled forward (F×D) and backward (R×D) was calculated. The start of the movement was defined as the first frame, the barbell moved vertically, and the end of the movement was defined as the first frame the vertical position of the barbell ceased to move downwards [46]. This captured using the same camera set-up described above, capturing the trajectory of two retroreflective markers on the right–and left-hand side of the barbell.

## Movement criterion: Barbell trajectory type

Overall barbell patterns were assessed using adapted criteria by Cunanan, Hornsby [42] of elite weightlifting trajectories. The following categories were implemented: type one–initial backward movement from the start, then away and being caught close to the center reference line; type two–backward movement from the start position, and does not cross the vertical reference line during movement; type three–away movement from the start position followed by toward and then away from the body; and type four–classified as a beginner trajectory, capturing movements that do not adhere to the specifications of the preceding categories. Barbell trajectories were categorized using extracted X Y coordinate data normalized to 100 data points. The summed frequency of each trajectory was used for further analysis.

## Data analysis

### Statistical analysis: Performance accuracy, movement criterion, imagery ability

A 3 (technique assessments: 1, 2, 3) × 2 (condition: NLP and LP group) factorial design was used to assess performance accuracy scores. Following the assessment of normality and homogeneity of variance, a mixed-design ANOVA was used to determine difference within and between groups for two dependent variables: F×D and R×D. When violations of sphericity were detected, $p$ values were corrected using Greenhouse-Geisser epsilon ($\varepsilon$) correction when mean epsilon was less than .75 and Hyun-Feld when mean epsilon was greater than 0.75. Post Hoc tests were implemented with Bonferroni correction applied to analyse significant main effects and interactions to determine the location of differences within (technique assessment) and between (conditions) factors, with statistical differences accepted at $p < .05$. A one-way ANOVA was used to examine baseline differences in movement imagery ability between the two conditions for the combined visual and kinaesthetic imagery scores. Partial eta squared ($\eta_p^2$) was used to express the magnitude of effects and interpreted as: small 0.02; medium 0.13; and large 0.26 [47]. Mann-Whitney tests were used to compute pair-wise comparisons for independent samples when normality and/or homogeneity of variance was not observed. Chi-square test-for-independence was conducted to analyse whether the frequency of trajectory type was related to the condition, and Bonferroni corrected $z$–tests to compare differences in trajectory frequency between conditions and technique assessments [48].

### Cluster analysis: Quantifying movement patterns exhibited

The number of different movement patterns demonstrated by each participant was quantified using a cluster analysis technique [29]. The cluster analysis was calculated by establishing one

time series of each trial (normalized to 100 data points), participants, technique assessments, and conditions. Subsequently, this cluster analysis method allows all trials to be grouped into meaningful clusters, where the number of 'actual' clusters is not known a priori. An iterative cluster algorithm (Fisher-EM) was utilised [49]. The Fisher-EM algorithm projects data into a new subspace for each iteration so that clusters emerging from the data set maximise the inter-cluster distance while minimising the intra-cluster distance [49]. This method enabled the identification of variability present in practice (i.e., number of movement patterns) and whether participants engaged in exploration of the movement, evidenced by high switching between movement patterns trial to trial [29].

### Exploratory and exploitative behaviour

Building on previous MI research, the present study aimed to examine exploratory and exploitive behaviours exhibited during LP and NLP forms of MI practice. Therefore, the number of movement patterns visited by each participant was calculated to show the number of different coordination patterns explored across the intervention. A coordination cluster was defined as being visited when displayed at least once throughout the three technique assessments. Furthermore, to determine exploratory and exploitive behaviours, all trials with the associated coordination cluster were plotted in chronological order. Exploitation was demonstrated when the same cluster was displayed in two consecutive trials. Exploration was defined as different movement patterns exhibited in two consecutive trials [29]. To establish whether participants engaged in more exploratory or exploitive behaviour, an exploration/exploitation ratio (EER) was calculated by dividing the number of exploration behaviours by the number of exploitation behaviours. Based on similar research by Komar, Potdevin [29], an EER of 1 denotes a balance between exploratory and exploitative behaviours, whereas a high EER (e.g., 1.5) indicates more significant levels of exploration. In the present study, the EER was implemented to examine potential differences in exploration and exploitation.

## Results

### Imagery ability

The analysis of general imagery ability showed that there was no main effect of condition for kinesthetic imagery score, $F(1,13) = .530$, $p = .480$ (NLP: 5.14 ± 1.07; LP: 4.68 ± 1.30), and the combined score $F(1,13) = 2.718$, $p = .125$ (NLP: 5.61 ± 0.73; LP: 4.68 ± 1.30). There was a significant main effect of condition for visual imagery score $F(1,13) = 5.438$, $p = .038$ (NLP: 6.07 ± 0.73; LP: 4.68 ± 1.40). However, both groups were above the acceptable average of 4 (neutral, not easy not hard) and were considered to have adequate MI ability [16].

### Movement patterns: Coordination profiling

From a potential 630 trials, 604 were successfully reconstructed for further analysis. For a record of relevant raw data, see S2 File. According to the Bayesian information criterion (BIC) indicator, the model that was most representative of the present data set revealed 11 emerging movement patterns across the 9 kinematic joint variables for the 3 technique assessment sessions (Fig 2). The BIC values for 2 to 20 potential patterns indicated that the values for 11 patterns were the beginning of the plateau of BIC values* [BIC values for 2 to 20 potential patterns respectively = -1064006; -1040288; -1022128; -1007447; -995841.9; -985810.5; -978462.3; -9695021.4; -956089.7; **-944909**\*; -941363.4; -931305.9; -922973.3; -915530.7; -911633.4; -905570.3; -2904090.5; -897949.6; -894890.9].

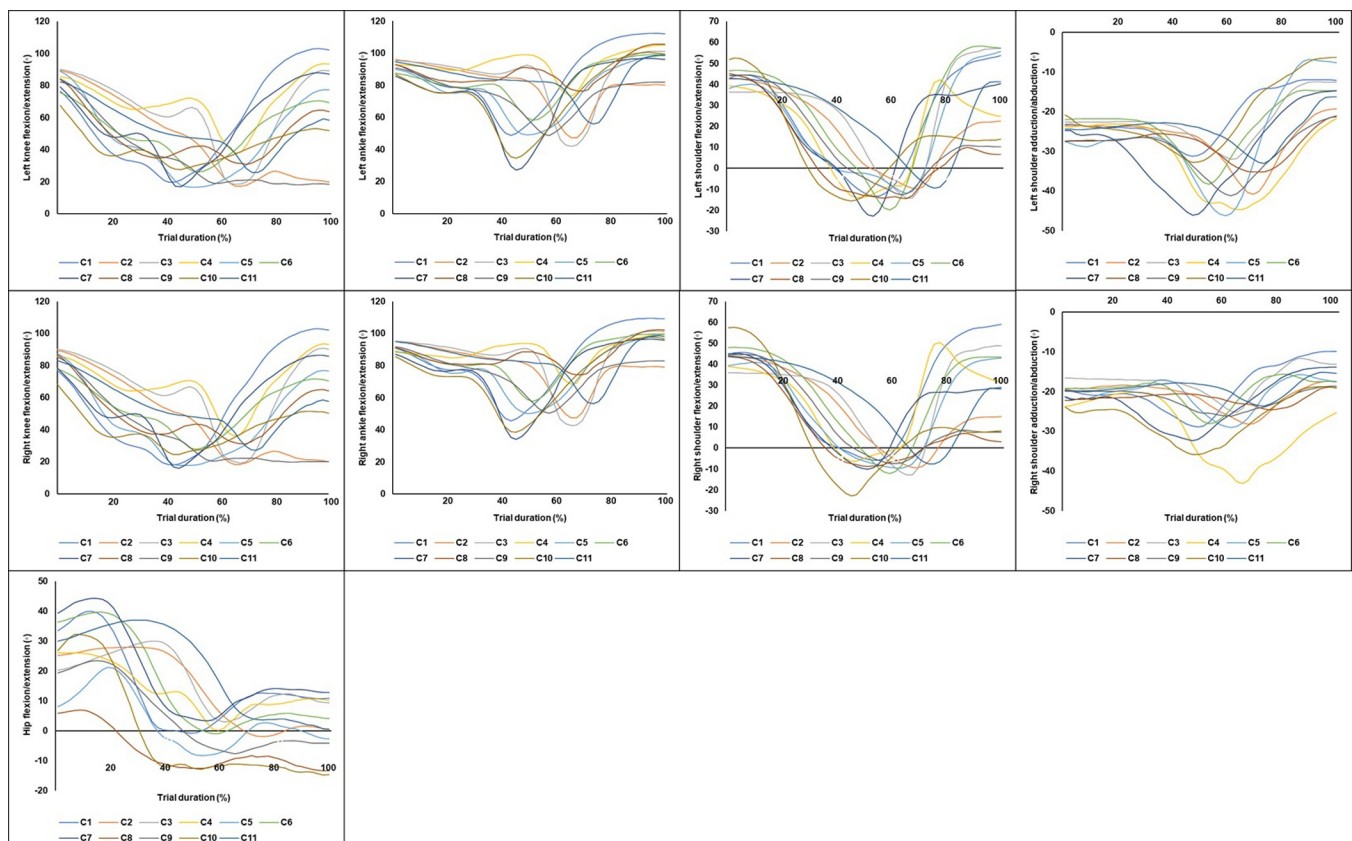

**Fig 2. Mean movement patterns normalized to 100 data points for each cluster within all kinematic variables across all technique assessment sessions for NLP and LP conditions.**

### Distribution of movement patterns

Across all three technique assessment sessions the NLP condition (C2, C3, C5, C6, C10) and LP condition exhibited five preferred patterns (C1, C4, C8, C9, C11). The distribution of trials for each movement pattern within technique assessments 1, 2, and 3 are shown in Fig 3. It was found that C5 and C6 comprised the largest number of trials for the NLP condition (15% and 28%, respectively). C5 was found to be a unique movement to participant NLP 10, with 100% of trials utilizing this movement pattern, suggesting strong initial behavioral tendencies that the task constraints could not successfully perturb. C11 and C4 comprised the highest number of trials (17% and 14%, respectively) for the LP condition. Similarly, C4 was only displayed by LP2 and comprised of 100% of trials, indicating no exploration.

Across all trials and both conditions, it was found that C6 comprised the largest number of movements (37% of total trials; NLP = 28%; LP = 8%). C6 was found to have the lowest F⊆D and R⊆D, indicating a more effective movement pattern as the barbell did not travel excessively away from the learner's body (F⊆D = 0.07 ± 0.04m) and the barbell ended in a more stable position near the learner's base of support (R⊆D = 0.06 ± 0.04m). Fig 3 shows that 44% (NLP = 27%; LP = 17%) of movements were belonged to C6 in technique assessment 3.

Fig 4A–4D displays individual time series plots for a representative NLP and LP participant's sample. Four primary exploration/exploitation patterns were observed, with two being shared by both conditions (Fig 4C). Pattern A (Fig 4A) displayed by the LP condition was characterized by exploitation early in learning (technique assessment 1 & 2), concluding with increased

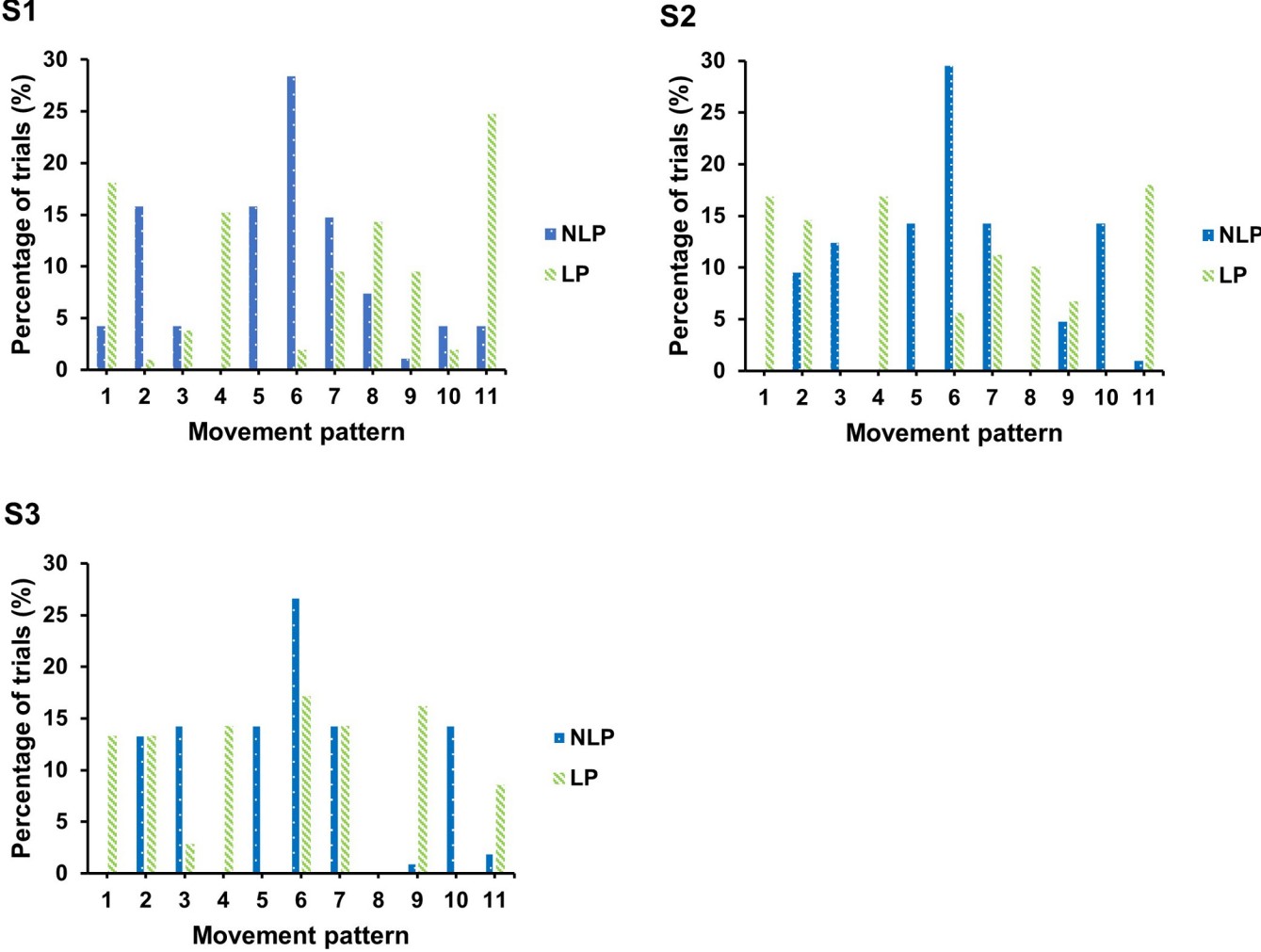

**Fig 3. Percentage of trials for movement patterns in each technique assessment session.** C6 comprised the most trials from session 1 (28%) to test 3 (27%) for the NLP condition. For the LP condition, C11 displayed the greater number of trials (25%) in session 1 and decreased in session 3 (9%), with C6 being the highest frequency movement in the final session (17%).

exploration late in learning. For example, LP6 initially exploited C11 for 30 consecutive trials before exploring three new movement patterns (C1, C3 and C6). Pattern B (Fig 4B) displayed by the NLP condition was included exploration early in learning, followed by increased exploitation. NLP14 demonstrated pattern B, exploring four-movement patterns (C8, C9, C10 and C11) between trials 0–10, followed by exploitation of C10 from trials 13–45. Pattern C (Fig 4C) was shared by both conditions and was characterized by early exploitation of a movement pattern that served as a platform for brief periods of exploration and returned to the initially exploited pattern early in practice. For example, LP7 exploited C7 after 4 trials and subsequently explored C1 and C6 in technique assessment 2 before returning to C7 in technique assessment 3. Finally, Pattern D (Fig 4D) was demonstrated by both conditions and was characterized by no exploration, with participants completely exploiting one movement pattern.

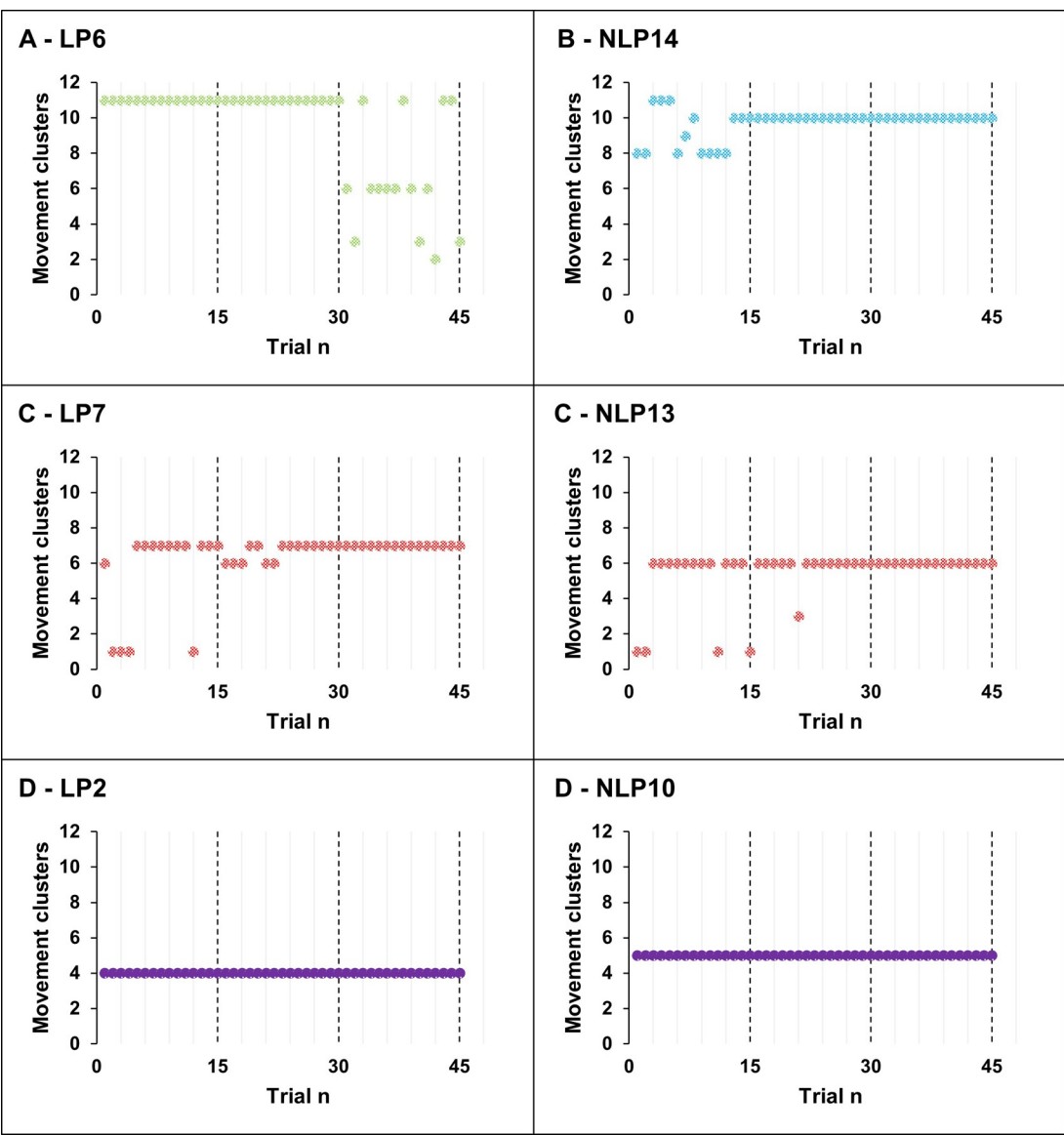

**Fig 4.** A, B: Time series plot of participants displaying exploration/exploitation pattern A (LP6) and B (NLP14) across technique assessment session 1, 2, and 3. Vertical dashed lines indicate the conclusion of each session. C, D: Time series plots of shared exploration/exploitation patterns C and D.

## Performance accuracy: Horizontal bar displacement

From a potential 630 trials, 604 were successfully reconstructed for performance accuracy analysis. The two-way ANOVA revealed a significant interaction effect on rearward barbell displacement (R⊆D) between condition and time of technique assessment, $F_{(2, 22)} = 5.040$, $p = .03$, $\eta_p^2 = .292$. Examination of means indicated that although there was a decrease in R⊆D for the NLP group from technique assessment 1 (0.06±0.43m) to 3 (0.05±0.32m), the LP condition showed an increase in R⊆D from technique assessment 1 (0.07±0.05m) to 3 (0.10 ±0.07m). Bonferroni post-hoc tests showed no significant differences between groups for technique assessments 1, 2, or 3 ($p = .13 - .67$). The main effects of time of technique assessment ($F_{(2, 22)} = 1.03$, $p = .373$ $\eta_p^2 = .086$) and condition ($F_{(1, 11)} = 0.78$, $p = .39$, $\eta_p^2 = .067$), were not

significant, respectively. Further analysis showed that for F⊆D the main effects of time of technique assessment ($F(2, 22) = 1.31$, $p = .29$, $\eta_p^2 = .106$) and condition ($F(1, 11) = 0.22$, $p = .64$, $\eta_p^2 = .020$), were not significant, respectively. The time of technique assessment ⊆ condition interaction was not significant for F ×D ($F(2, 22) = .157$, $p = .856$, $\eta_p^2 = .014$).

## Movement criterion: Barbell trajectory type

Fig 5 shows examples of each barbell trajectory from representative participants. For the NLP condition, 72% of total trials were type 3 trajectories, 27% were type 4, and 2% were type 1 (criterion model). In the LP condition, 54% of total trials were type 3 trajectories, 23% were type 4, 22% were type 1 (criterion model), and 1% were type 2 trajectories. The NLP condition did not display type 2 trajectories and was therefore not included for further analyses. A Pearson's chi-square test of independence was used to evaluate whether barbell trajectory type was related to condition (NLP or LP). The chi-square test was statistically significant, $\chi^2(2, n = 586) = 56.311$, $p < .001$, Cramer's $V = .310$, indicating a moderate association. $Z$–tests with Bonferroni correction revealed that the frequency of type 1 trajectories was significantly higher for the LP condition (22% of trials) compared to the NLP condition (2% of trials) ($z = 5.1$, $p < .001$, two-tailed). A significant difference was detected for type 3 trajectories between the NLP (72% of trials) and LP condition (57% of trials) ($z = -1.6$, $p < .05$, two-tailed). Further analysis revealed that the frequency of type 3 trajectories was significantly greater than type 1 trajectories in both the LP ($z = -2.0$, $p < .05$) and NLP conditions ($z = 2.0$, $p < .05$).

## Movement pattern: Visited and exploited

Table 2 displays the number of movement patterns visited and exploited by each participant. Mann -Whitney tests revealed no significant differences in visited ($p = .315$) and exploited number of patterns ($p = .165$) between conditions.

## Exploratory and exploitative behaviour: Exploration/exploitation ratio

The exploration/exploitation ratio was calculated to examine how two different approaches to skill acquisition may influence subsequent modes of behaviour following MI practice. No significant difference was found in the number of exploratory and exploitive behaviour between the LP and NLP conditions ($p = .438$; Table 3).

## Impact of constraints on individual exploratory behaviour

Examination of EER indicates that the incorporation of constraints in the MI instructions of the NLP condition had a distinctly different impact on the individual exploratory behaviours as a function of time. For example, NLP9, NLP13, NLP14 and NLP16 displayed their peak EER in technique assessment 1 (EER = 0.25–4) and their lowest EER in technique assessment 2 after a period of MI practice with constraints present (EER = 0–0.67). By contrast, NLP5 and NLP12 displayed their highest EER in technique assessment 2 (EER = 1.5–3). NLP10 displayed complete exploitation (EER = 0) across all three assessments.

## Discussion

The purpose of the present study was to investigate the application of a NLP informed MI approach to skill acquisition. Our aim was not to propose the extent of effectiveness for NLP informed MI. Rather, the goal of the present study was to provide preliminary findings to stimulate discussion around alternative pedagogical approaches to MI for developing motor skills. Consistent with hypothesis (1), the LP condition displayed a higher frequency of the

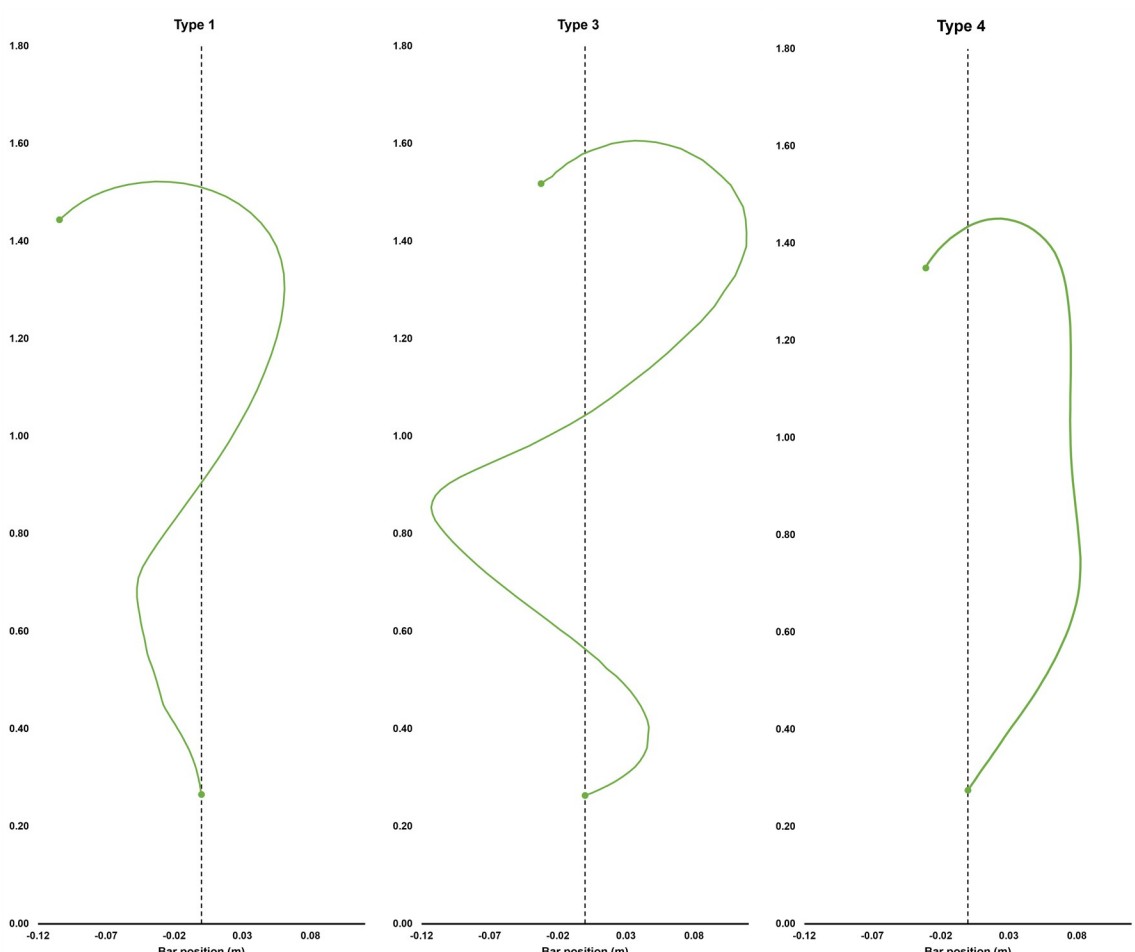

**Fig 5. Example of each demonstrated barbell trajectory from three individual learners.** Each barbell trajectory was defined by its relationship to the vertical reference line (Cunanan et al., 2020). Each barbell trajectory was normalised to 100 data points and plotted using the Y (horizontal), and Z (vertical) coordinates extracted from visual 3-D software. Type 2 trajectory was not exhibited by either condition.

prescribed technical model (i.e., type one trajectory) than the NLP condition. However, in both conditions, the prevalence of type 3 trajectories was significantly greater than any other barbell trajectory. Partially consistent with hypotheses (2) and (3), exploration was observed in the NLP condition but not significantly more than LP. Practice and performance accuracy (R⊆D) improved equally for both conditions.

Type one barbell trajectories are touted as the most efficient and 'optimal' technique for weightlifting movements [41]. Research, however, suggests this particular trajectory is typically

**Table 2. The number of different movement patterns visited during three testing sessions and the number of movement patterns exploited between at least two consecutive trials.**

| | Number of movement patterns explored | | | | | | | | Number of movement patterns exploited | | | | | | | |
|---|---|---|---|---|---|---|---|---|---|---|---|---|---|---|---|---|
| | Participants | | | | | | | Mean | Participants | | | | | | | Mean |
| Condition | 1 | 2 | 3 | 4 | 5 | 6 | 7 | | 1 | 2 | 3 | 4 | 5 | 6 | 7 | |
| NLP | 2 | 2 | 1 | 2 | 3 | 4 | 5 | 2.29 | 1 | 1 | 1 | 2 | 1 | 2 | 1 | 1.29 |
| LP | 4 | 1 | 3 | 2 | 4 | 6 | 2 | 3.14 | 3 | 1 | 2 | 1 | 2 | 2 | 2 | 1.86 |

**Table 3. Exploration/exploitation ratio for each participant for the NLP and LP conditions.**

| | Participants | | | | | | | |
|---|---|---|---|---|---|---|---|---|
| Condition | 1 | 2 | 3 | 4 | 5 | 6 | 7 | Mean |
| NLP | 0.52 | 0.22 | 0 | 0.27 | 0.28 | 0.51 | 0.06 | 0.26 |
| LP | 0.65 | 0 | 0.30 | 0.07 | 0.94 | 0.89 | 0.03 | 0.41 |

not the most utilised in elite competitions [42]. Cunanan, Hornsby [42] observed that type 1 trajectories were displayed the least in both male (12% of lifts) and female lifters (12% of lifts) across all weight categories at the 2015 world weightlifting championships. Interestingly, despite supposed inefficiency, type 3 trajectories were demonstrated most frequently in males (51% of lifts) and females (56% of lifts). A similar pattern was observed in the present study in beginner level lifters with both the LP and NLP conditions demonstrating a significantly higher proportion of type 3 barbell trajectories than type one, despite the LP condition being explicitly instructed to perform a type one trajectory. These findings indicate that the 'optimal' technique may not be constrained to a particular barbell trajectory (i.e., type one barbell trajectory). The higher prevalence of type 3 trajectories for the LP condition suggests that regardless of what technical model is taught, individual movement constraints may require learners to search for coordination patterns that align with their capabilities and meet the task's demands [30]. Previous MI research appears consistent with this contention, with Lindsay, Oldham [27] reporting that PC technique was highly individualised in novice lifters following 6 weeks of MI practice, regardless of instructional approach (i.e., prescriptive or personalised).

One evident issue with comparing the present findings with observational data from elite performers is the obvious difference in overall performance outcome. The utilisation of 'sub-optimal' technique can be easily justified in elite performers when such high-performance levels are attained [42]. Subsequently, the demonstration of sub-optimal technique is frequently viewed by practitioners to impede the development of expertise [26]. Performance accuracy scores from the present study suggest that exploration of movement patterns that deviate from an 'optimal' technical model do not necessarily impede performance and may be an essential part of developing skilled behaviour. Participants in both the NLP and LP conditions demonstrated a preference for distinctly different coordination patterns. Within the LP condition, participants primarily exhibited C1, C4, C8, C9, and C11. By contrast, the incorporation of task constraints in the NLP condition limited the expression of these patterns, preferring C2, C3, C5, C6, and C10. Despite the use of distinctly different coordination patterns, improvements in R⊆D was the same for both groups, suggesting that more than one technique can produce the same overall performance outcome. These findings imply that the focus of MI practice for skill acquisition may be to facilitate learners in their search for individually appropriate coordination patterns rather than prescribe a specific way of performing a skill. If learners are inclined to deviate from the instructed technique, MI practice that allows exploration of less 'optimal' movements may provide necessary opportunities for learners to develop individually appropriate coordination patterns. As demonstrated in the present study, effective movement may not adhere to a prescribed technique. Instead, effective movement can be expressed as movement organisation that meets constraints of the perceptual-motor workspace while attaining improved performance [30]. Subsequently, MI practitioners may want to consider implementing analogy or movement outcome focused instructions, rather than defining an explicit movement model [8, 30].

Regarding the quantity of exploration, no significant differences were observed between LP and NLP conditions, suggesting that the constraints incorporated into the MI scripts were not a precondition for exploration. One potential explanation is that exploratory behaviour in the LP

condition demonstrates participants attempting to follow the prescribed movement but may not have been individualised to their constraints (i.e., limb length and body weight) [50]. Exploration in the LP condition might be defined more appropriately as coordination instability where participants are 'caught' between an inherent self-organising process and the need to conform to a specific movement pattern. Whereas practice for NLP participants introduced variability to encourage exploration and the emergence of individualised movement solutions [21].

Examination of exploratory behaviour over time indicated that responses to constraints was highly individualised for NLP participants. NLP5 and NLP12 increased exploration between technique assessment 1, when constraints were absent (EER = 1 & 0) and assessment 2 after a period of practice with constraints present (EER = 3 & 1.5). Conversely, NLP9, NLP13, NLP14, NLP16 constraints appeared to restrict exploratory behaviour, with EER decreasing between technique assessment 1 (EER = 0.25–4) and 2 (0–0.67). Furthermore, when examining the distribution of trials for each preferred coordination cluster. NLP participants demonstrated a preference for C6, which was found to be the most 'effective' cluster displaying limited barbell movement away from the learner's body (F$\subseteq$D = 0.07 ± 0.04m) and the barbell ended in a more stable position near the learner's base of support (R$\subseteq$D = 0.06 ± 0.04m). These findings indicate that exploration quantity is not necessarily the determining characteristic for developing individually relevant coordination patterns. Instead, the important point is that the nature of exploration elicited by the constraints was functionally relevant for each individual, resulting in optimal task solutions (i.e., improved R$\subseteq$D). In support of these findings, Barris et al. [51] observed in elite springboard divers that coordination patterns variability increased and decreased after practice utilizing variable take-off conditions, but performance outcomes improved under all conditions.

The individualized nature of exploration highlights an important limitation of the current study. The EER can only provide a measure of exploration/exploitation quantity and does not shed light on the nature of exploration for each learner, which the present study shows may be distinctly varied. This raises questions about what can be considered an optimal level of exploration. Exploration can be described as a process of attuning to reliable information throughout the movement, suggesting that the effectiveness of exploratory behaviour cannot be solely attributed to an increase in the amount of exploration [23, 29, 50]. Rather, effective exploration may be an improved ability to attune to opportunities for action that align with the learners' capabilities and experiences [23, 50]. Therefore, further research should consider how learners perceive information during MI practice when investigating exploratory behaviour. The individualized nature of exploration in the present study raises an important issue for introducing variability into skill practice using MI. It is possible that the constraints could not effectively perturb initially strong behavioural tendencies leading to a reduction in exploration. Therefore, to effectively facilitate exploration, constraints in MI instructions could be adjusted throughout practice more regularly. Although the initial presentation of constraints may have encouraged exploration, these constraints may have become 'outdated' and needed to be changed to perturb newly stabilised coordination patterns. This is in line with the idea that constraints can emerge and decay over time or with learning [30, 50]. The layered stimulus response approach to MI instructions is consistent with this idea, where script information is gradually layered over time [52]. A fruitful line of inquiry may be to examine the influence of gradually adjusting or removing constraints over time to challenge individual coordination patterns.

## Conclusion

In summary, the present study provides preliminary findings on applying NLP principles of skill acquisition in MI practice. Similar quantities of exploration were observed in both

conditions, indicating that regardless of instructions, learners may inherently explore opportunities for action that align with individual capabilities and information presented in the MI practice environment. The utilization of 'sub-optimal' techniques by both conditions (i.e., type 3 trajectories) coupled with equivalent improvements in performance accuracy (i.e., rearward barbell trajectory) indicated that adhering to an 'optimal' technique does not ensure improved performance. These findings suggest that a movement's overall effectiveness for meeting the task goal may be of more importance than replicating a movement that looks correct in MI. The present study highlights the potential benefits of utilising a NLP approach to MI to encourage learners to explore movement solutions that align with a learner's capabilities without negatively impacting performance. It may be beneficial for MI practitioners to consider designing practice that allows deviations from prescribed technical models to facilitate learners' inherent exploration of individual task solutions. Future research should investigate further the efficacy of NLP informed MI to develop further understanding around how best to apply these principles of skill acquisition in MI practice.

## Supporting information

**S1 File. MI instructions for NLP and LP conditions.**
(PDF)

**S2 File. Relevant raw outcome data.**
(ZIP)

## Author Contributions

**Conceptualization:** Riki S. Lindsay, Jia Yi Chow, Paul Larkin, Michael Spittle.

**Data curation:** Riki S. Lindsay.

**Formal analysis:** Riki S. Lindsay, John Komar.

**Investigation:** Riki S. Lindsay, Jia Yi Chow, Paul Larkin, Michael Spittle.

**Methodology:** Riki S. Lindsay, John Komar, Jia Yi Chow, Paul Larkin, Michael Spittle.

**Project administration:** Riki S. Lindsay, Michael Spittle.

**Resources:** Michael Spittle.

**Software:** John Komar.

**Supervision:** Jia Yi Chow, Paul Larkin, Michael Spittle.

**Visualization:** Riki S. Lindsay, John Komar, Jia Yi Chow, Michael Spittle.

**Writing – original draft:** Riki S. Lindsay, John Komar, Jia Yi Chow, Michael Spittle.

**Writing – review & editing:** Riki S. Lindsay, John Komar, Jia Yi Chow, Paul Larkin, Michael Spittle.

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
