## [Decision Letter · Decision Letter 0]

2 Jan 2023

PONE-D-22-02521Characterising exploratory behaviour during motor imagery practice: a nonlinear pedagogy approachPLOS ONE

Dear Dr. Lindsay,

Thank you for submitting your manuscript to PLOS ONE. After careful consideration, we feel that it has merit but does not fully meet PLOS ONE’s publication criteria as it currently stands. Therefore, we invite you to submit a revised version of the manuscript that addresses the points raised during the review process.

ACADEMIC EDITOR.Dear Authors, one expert in the field reviewed your manuscript retrieving several major issues you should consider in the revision process. Please submit your revised manuscript by Feb 16 2023 11:59PM. If you will need more time than this to complete your revisions, please reply to this message or contact the journal office at plosone@plos.org. Please include the following items when submitting your revised manuscript:A rebuttal letter that responds to each point raised by the academic editor and reviewer(s). You should upload this letter as a separate file labeled 'Response to Reviewers'.A marked-up copy of your manuscript that highlights changes made to the original version. You should upload this as a separate file labeled 'Revised Manuscript with Track Changes'.An unmarked version of your revised paper without tracked changes. You should upload this as a separate file labeled 'Manuscript'.

We look forward to receiving your revised manuscript.

Kind regards,

Emiliano Cè

Academic Editor

PLOS ONE

and https://journals.plos.org/plosone/s/file?id=ba62/PLOSOne_formatting_sample_title_authors_affiliations.pdf.

Reviewers' comments:

Reviewer's Responses to Questions

**Comments to the Author**

1. Is the manuscript technically sound, and do the data support the conclusions?

Reviewer #1: Yes

2. Has the statistical analysis been performed appropriately and rigorously? 

Reviewer #1: Yes

3. Have the authors made all data underlying the findings in their manuscript fully available?

Reviewer #1: Yes

4. Is the manuscript presented in an intelligible fashion and written in standard English?

Reviewer #1: Yes

5. Review Comments to the Author

Reviewer #1: This paper proposes applying nonlinear pedagogy (NLP) principles of skill acquisition in motor imagery (MI) practice, which demonstrate that careful and considered manipulation of task constraints can leverage movement variability (exploration) to facilitate the adoption of individualised movement solutions. In this study, fourteen weightlifting beginners (2 female and 12 male) participated in a 4-week intervention involving either NLP (i.e. analogy-based instructions and manipulation of task constraints) or a linear pedagogy (LP; prescriptive instructions of optimal technique, repetition of same movement form) to learn a complex weightlifting derivative. Similar results were observed in both conditions, indicating that a movement's overall effectiveness for meeting the task goal may be of more importance than replicating a movement that looks correct in MI. It may be beneficial for MI practitioners to consider designing practice that allows deviations from prescribed technical models to facilitate learners' inherent exploration of individual task solutions. Some related studies could be discussed: Here are some questions that should be answered: Robust Similarity Measurement Based on a Novel Time Filter for SSVEPs Detection，2021，IEEE Transactions on Neural Networks and Learning Systems，2021,DOI: 10.1109/TNNLS.2021.3118468, Correlation-based channel selection and regularized feature optimization for MI-based BCI，Neural Networks，2019,118:262-270, An Asynchronous Hybrid Spelling Approach Based on EEG-EOG Signals for Chinese Character Input，IEEE TRANSACTIONS ON NEURAL SYSTEMS AND REHABILITATION ENGINEERING， 2019， 27（6）：1292-1302

1. Page 2, lines 21-22. “However, little research has investigated the how alternative approaches to skill acquisition, such as nonlinear pedagogy (NLP), can be applied to MI. ” Is there an extra ‘how’ here?

2. Page 2, line 27. It's better to replace ‘beginner weightliters’ with ‘weightlifting beginners’.

3. Abstract. It is better to illustrate the results with concrete figures.

4. Some statements in the article seem to have grammatical problems. Please check and modify.

5. Page 5, line 98. Can you use an example of Nonlinear Pedagogy (NLP) to explain more specifically what it means?

6. Page 7, line 138. “MI scripts are the only viable way to ‘manipulate’ task constraints, given that practice is performed in the mind.” Is there any research to support the claim of 'only'?

7. Page 7, line 150. What does ‘F×D’ and ‘R×D’ mean?

8. Page 8, lines 175-177. Is the word 'consisitency' missing after 'high inernal'?

9. Pages 8-9, lines 178-187. It will be easier to understand to use a figure to represent anatomical landmarks.

10. Pages 11, lines 236-238. Why introduce constraints to the NLP condition on sessions 3-6?

11. Can you list NLP and LP scripts?

12. Pages 14, line 283. How are the coordinates of X and Y set?

13. Pages 16, line 331. There is a lack of content here.

14. Figure 1. What does ‘rep duration’ mean?

6. PLOS authors have the option to publish the peer review history of their article (what does this mean?). If published, this will include your full peer review and any attached files.

Reviewer #1: No

---

## [Author Response · Author response to Decision Letter 0]

15 Feb 2023

Response to Reviewers

Comment 1: Some related studies could be discussed: Here are some questions that should be answered: Robust Similarity Measurement Based on a Novel Time Filter for SSVEPs Detection，2021，IEEE Transactions on Neural Networks and Learning Systems，2021,DOI: 10.1109/TNNLS.2021.3118468, Correlation-based channel selection and regularized feature optimization for MI-based BCI，Neural Networks，2019,118:262-270, An Asynchronous Hybrid Spelling Approach Based on EEG-EOG Signals for Chinese Character Input，IEEE TRANSACTIONS ON NEURAL SYSTEMS AND REHABILITATION ENGINEERING， 2019， 27（6）：1292-1302

Response 1: Thank you for your detailed review of our manuscript. Upon thoroughly reading the suggested works, it is not clear to us that the articles would provide support, from either a methodological or theoretical perspective, to the current study. If the reviewer is able to provide some further justification, or clarification on the specifics from these studies that are applicable to our project, we would be more than happy to consider. However, in the meantime, we have not made any modifications in relation to this in the manuscript. 

Comment 2: 1. Page 2, lines 21-22. “However, little research has investigated the how alternative approaches to skill acquisition, such as nonlinear pedagogy (NLP), can be applied to MI. ” Is there an extra ‘how’ here?

Response 2: Thank you for raising this point. We have amended this section as recommended. (Page 2; Lines 25 - 26)

Comment 3: Page 2, line 27. It's better to replace ‘beginner weightliters’ with ‘weightlifting beginners’.

Response 3: Good point. We have changed this section to read as :weightlifting beginners” (Page 2; Line 33).

Comment 4: 3. Abstract. It is better to illustrate the results with concrete figures.

Response 4: Agreed. Thank you for pointing this out. We have substantially edited the abstract to include specific figures to read as follows (Page 2, Lines 38 – 46):

“No significant differences (p = .438) were observed in the amount of exploration between LP (EER = 0.41) and NLP (EER = 0.26) conditions. Equivalent changes in rearward displacement (R×D) were observed with no significant differences between conditions for technique assessments 1, 2, or 3 (p = .13 - .67). Both NLP and LP conditions were found to primarily demonstrate ‘sub-optimal’ type 3 barbell trajectories (NLP = 72%; LP = 54%). These results suggest that MI instructions prescribing a specific movement form (i.e., LP condition) are ineffective in restricting available movements to a prescribed technique but rather the inherent task constraints appear to ‘force’ learners to explore alternative movement solutions to achieve successful performance outcomes.”

Comment 5: 4. Some statements in the article seem to have grammatical problems. Please check and modify.

Response 5: Article as been thoroughly checked and editing software has been utilised to ensure no grammatical errors.

Comment 6: 5. Page 5, line 98. Can you use an example of Nonlinear Pedagogy (NLP) to explain more specifically what it means?

Response 6: This section has been edited to incorporate an example from the literature to clarify NLP further (Page 4, Lines 89-98):

“Contemporary approaches to motor learning, such as Nonlinear Pedagogy (NLP), draw on the ecological dynamics view to provide a skill development framework that acknowledges the emergent nature of skilled action. From a NLP perspective, the aim is to develop an individual’s ability (i.e., performer constraints) to successfully adapt their movements to changing environmental constraints. Therefore, behavioural flexibility is a key attribute of skilled action, in which an individual can explore possible movements and attune to relevant information in satisfying the overall task goal (Komar et al., 2023; Rangathan et al., 2020). Individuals adapt to environmental perturbations by exploring alternative movement options to meet the changing task demands. For example, Muller et al. (2015) showed that expert cricket batsmen practicing under occluded and normal vision conditions demonstrated unique coordination patterns, yet performance outcomes were equivalent between all batsmen. These findings indicate that movement variability, though commonly viewed as detrimental to performance, could play a functional role in facilitating the development of individualised performance solutions.”

Comment 7: Page 7, line 138. “MI scripts are the only viable way to ‘manipulate’ task constraints, given that practice is performed in the mind.” Is there any research to support the claim of 'only'?

Response 7: This section has been adjusted to clarify this statement and provide support.

Comment 8: Page 7, line 150. What does ‘F×D’ and ‘R×D’ mean?

Response 8: This statement has been amended to clarify these terms to read as follows (Page 7, Lines 147-148):

“as measured by the distance the barbell travels forward (F x D) and backward (R x D) relative to the start position.”

Additionally, further detail on these measures is provided under the “Performance accuracy: horizontal barbell displacement” section (page 13, lines 261-267).

Comment 9: Page 8, lines 175-177. Is the word 'consisitency' missing after 'high inernal'?

Response 9: This has been corrected.

Comment 10: Pages 8-9, lines 178-187. It will be easier to understand to use a figure to represent anatomical landmarks.

Response 10: Agreed. A figure 1 has been provided to visually represent anatomical landmarks.

Comment 11: Pages 11, lines 236-238. Why introduce constraints to the NLP condition on sessions 3-6?

Response 11: Thank you for raising this point. Constraints were introduced for these sessions for a number of reasons. Firstly, variability/exploration is a key principle of NLP learning design and constraint manipulation is the means in which to leverage variability in practice. Secondly, constraints were introduced for these sessions to reduce cognitive overload for participants in the early stages of learning. Finally, though constraints were physically present for the NLP condition, this was only as a visual aid for MI. NLP participants never physically practiced with constraints, therefore, physical testing sessions were identical for LP and NLP conditions. The relevant section has been amended to read as follows (Page 11, Lines 231-237):

“The introduction of practice variability to encourage learner exploration is a key principle of NLP learning design (30). Subsequently, constraints were described in MI instructions for the NLP condition to introduce participants to variability. In order not to overload participants in the early stages of learning, variability was introduced into MI practice between sessions 3 – 6 (26). To replicate the physical practice environment, constraints were physically present for the NLP condition, however, learners only imagined practicing under constraints and these were not present for any testing sessions.”

Comment 12: Can you list NLP and LP scripts?

Response 12: Yes, scripts have been provided as supporting information.

Comment 13: Pages 14, line 283. How are the coordinates of X and Y set?

Response 13: X, Y and Z coordinates were established in Visual 3D by using a static calibration trial to establish locally fixed segment coordinate systems and establish the fixed A for each tracking target. Allowing for the motion of selected anatomical segments to be measured. This information has been included in the manuscript to clarify this point (Page 14, Lines 268-269):

“A static calibration trial was used to establish locally fixed segment coordinate systems and establish X, Y and Z axes.”

Comment 14: Pages 16, line 331. There is a lack of content here.

Response 14: Apologies, this has been amended.

Comment 15: Figure 1. What does ‘rep duration’ mean?

Response 15: This has been changed to read “Trial duration” for greater clarity.

---

## [Decision Letter · Decision Letter 1]

20 Feb 2023

Different pedagogical approaches to motor imagery both demonstrate individualized movement patterns to achieve improved performance outcomes when learning a complex motor skill

PONE-D-22-02521R1

Dear Dr. Lindsay,

We’re pleased to inform you that your manuscript has been judged scientifically suitable for publication and will be formally accepted for publication once it meets all outstanding technical requirements.

Kind regards,

Emiliano Cè

Academic Editor

PLOS ONE

Additional Editor Comments (optional):

Reviewers' comments:

Reviewer's Responses to Questions

**Comments to the Author**

1. If the authors have adequately addressed your comments raised in a previous round of review and you feel that this manuscript is now acceptable for publication, you may indicate that here to bypass the “Comments to the Author” section, enter your conflict of interest statement in the “Confidential to Editor” section, and submit your "Accept" recommendation.

Reviewer #1: All comments have been addressed

2. Is the manuscript technically sound, and do the data support the conclusions?

Reviewer #1: Yes

3. Has the statistical analysis been performed appropriately and rigorously? 

Reviewer #1: Yes

4. Have the authors made all data underlying the findings in their manuscript fully available?

Reviewer #1: Yes

5. Is the manuscript presented in an intelligible fashion and written in standard English?

Reviewer #1: Yes

6. Review Comments to the Author

Reviewer #1: This paper have been revised according to the comments. Authors have addressed my questions. This paper could be accepted.

7. PLOS authors have the option to publish the peer review history of their article (what does this mean?). If published, this will include your full peer review and any attached files.

Reviewer #1: No

---

## [Editor Report · Acceptance letter]

6 Mar 2023

PONE-D-22-02521R1 

Different pedagogical approaches to motor imagery both demonstrate individualized movement patterns to achieve improved performance outcomes when learning a complex motor skill 

Dear Dr. Lindsay:

I'm pleased to inform you that your manuscript has been deemed suitable for publication in PLOS ONE. Congratulations! Your manuscript is now with our production department. 

Kind regards, 

on behalf of

Professor Emiliano Cè 

Academic Editor

PLOS ONE